# Algorithms Trained on Normal Chest X-rays Can Predict Health Insurance Types

**Chi-Yu Chen**[1]                                                    ALTIS5526@NTUH.GOV.TW
[1] *National Taiwan University Hospital, Taiwan*
**Rawan Abulibdeh**[*,2]                                    RAWAN.ABULIBDEH@MAIL.UTORONTO.CA
[2] *University of Toronto, Canada*
**Arash Asgari**[*,3]                                                   ARASHASG@YORKU.CA
[3] *York University, Canada*
**Sebastián Andrés Cajas Ordóñez**[*,4]                                  SEBASMOS@MIT.EDU
[4] *MIT Critical Data, United States of America*
**Leo Anthony Celi**[*,5,6,7]                                              LCELI@MIT.EDU
[5] *Massachusetts Institute of Technology, United States of America*
[6] *Beth Israel Deaconess Medical Center, United States of America*
[7] *Harvard T.H. Chan School of Public Health, United States of America*
**Deirdre Goode**[*,8]                                                  DGOODE1@MGB.ORG
[8] *Mass General Brigham, Boston Massachusetts, United States of America*
**Hassan Hamidi**[*,3]                                                  HHAMIDI@YORKU.CA
**Ned McCague**[*,5,9]                                                  NMCCAGUE@MIT.EDU
[9] *Boston University, United States of America*
**Laleh Seyyed-Kalantari**[*,3,10]                                           LSK@YORKU.CA
[10] *Vector Institute, Canada*
**Thomas Sounack**[*,11]                                    THOMAS_SOUNACK@DFCI.HARVARD.EDU
[11] *Dana-Farber Cancer Institute, United States of America*
**Po-Chih Kuo**[12]                                              KUOPC@CS.NTHU.EDU.TW
[12] *National Tsing Hua University, Taiwan*

**Editors:** Accepted for publication at MIDL 2026

## Abstract

Artificial intelligence is revealing what medicine never intended to encode. Deep vision models, trained on chest X-rays, can now detect not only disease but also invisible traces of social inequality. In this study, we show that state-of-the-art architectures (DenseNet121, SwinV2-T, MedMamba) can predict a patient's health insurance type, a strong proxy for socioeconomic status, from normal chest X-rays with significant accuracy (AUC $\approx 0.70$ on MIMIC-CXR-JPG, 0.68 on CheXpert). The signal was unlikely contributed by demographic features by our machine learning study combining age, race, and sex labels to predict health insurance types. The signal also remains detectable when the model is trained exclusively on a single racial group. Patch-based occlusion reveals that the signal is diffuse rather than localized, embedded in the upper and mid-thoracic regions. This suggests that deep networks may be internalizing subtle traces of clinical environments, equipment differences, or care pathways; learning socioeconomic signals itself. These findings challenge the assumption that medical images are neutral biological data. By uncovering how models perceive and exploit these hidden social signatures, this work reframes fairness in medical AI: the goal is no longer only to balance datasets or adjust thresholds, but to interrogate and disentangle the social fingerprints embedded in clinical data itself. The codes are available at Link.

**Keywords:** spurious correlation, health insurance, medical images

---

* These authors contributed equally to this work; author names are placed in alphabetical order by last name.

## 1. Introduction

Artificial intelligence (AI) has become increasingly part of our everyday life. However, experts acknowledge concerns regarding AI fairness, specifically the disparate outcomes of an AI model across different societal subpopulations (Seyyed-Kalantari et al., 2020). Such considerations are paramount given that the outcomes of AI-facilitated decision-making are directly correlated with an individual's well-being. Recent studies (Geirhos et al., 2020; Kim et al., 2019; Buolamwini and Gebru, 2018) have demonstrated that deep learning methods are prone to exploit spurious correlation from shortcut features, which are predictive features that do not correspond to disease-specific patterns (Banerjee et al., 2023). The data distribution may also be polluted by multiple political, historical, or socioeconomic factors that exist in human society, containing unethical information that could be realized by AI algorithms and used by them. For example, (Obermeyer et al., 2019) showed that a widely used commercial health prediction system treated different racial groups unequally since black patients were likely to spend less on healthcare than white patients, which might be correlated to hindered access to healthcare facilities for black patients. Therefore, researchers have been investigating the equality of AI algorithms treating people with different demographic data attributes (such as sex, race, age) across various modalities. (Larrazabal et al., 2020; Kumar et al., 2025).

While shortcut features could be obvious and easy to identify, such as chest tube or watermarks on a medical image; we are more concerned with those that are imperceptible and unidentifiable to humans, such as demographic information in a chest X-ray. Demographic information could even introduce significant societal bias if they act as a shortcut feature. A leading study (Gichoya et al., 2022) showed that the race of patients is strongly detectable from chest X-rays and other studies localized race features on medical images to highlight how AI detect race from chest X-rays (Konate et al., 2025; Salvado et al., 2024). (Adleberg et al., 2022) discussed the hidden health insurance information in the chest X-ray images. However, the study included patients of all ages in MIMIC-IV, overlooking the fact that people older than 65 years old are automatically enrolled in public insurance. Moreover, they did not exclude the chest X-rays with thoracic diseases, which is another confounding factor of health insurance status.

In this finding paper, we demonstrated the potential of AI algorithms to detect health insurance types from chest X-ray images. The lower performance of AI model in Chest X-ray disease diagnosis (Seyyed-Kalantari et al., 2020) for low income patients with public insurance and the underdiagnosis bias against those patients has been documented in (Seyyed-Kalantari et al., 2021, 2020; Bahre et al., 2025), necessitating further investigation to validate the presence of health insurance information within chest X-ray images. Therefore, our contribution is as below:

1. To the best of our knowledge, this is the first in-depth analysis that reveals that deep vision models (DenseNet121 (Huang et al., 2017), SwinV2-T (Liu et al., 2022), Med-Mamba (Yue and Li, 2024)) can predict health insurance types from normal chest X-ray images (without thoracic diseases) in the MIMIC-CXR-JPG and CheXpert dataset. This suggests that chest X-rays (CXRs) contain latent socioeconomic information about patients that AI models can potentially leverage.

2. We demonstrated that these information distributed diffusely rather than localized and there is likely more information in the upper two-third part of a chest X-ray.

3. From close analysis between health insurance types and other demographic features (e.g. age, race, sex), we found that these common demographic features do not act as an essential mediator in our insurance type prediction task, which indicates that the deep vision models directly detect insurance type signals from chest X-ray images.

## 2. Methods

### 2.1. Datasets

In our experiment, two medical datasets were utilized: MIMIC-CXR-JPG (Johnson et al., 2019a,b) and CheXpert (Irvin et al., 2019). MIMIC-CXR-JPG is an extended image dataset for MIMIC-IV v3.0 (Johnson et al., 2024, 2023), including 377,110 chest X-ray images in total. We utilized MIMIC-CXR-JPG and linked it to MIMIC-IV v3.0 data by subject ID to attain patients' health insurance type. There are six health insurance categories in MIMIC-IV v3.0: Medicaid, Medicare, Private, Self-pay, No charge, and Other. On the other hand, CheXpert includes 224,316 chest X-rays from 65,240 patients from centers of Stanford Hospital. The recent CheXpert Plus paper (Chambon et al., 2024) provides additional health insurance information for each patient, in which there are three categories: Medicaid, Medicare, and Private. As for CheXpert, we utilized the JPG-downsized version due to computational limits. (See appendix A for detailed dataset descriptions.) For both datasets, we chose patients with Medicaid, Medicare, and Private insurance labels for analysis, further merging Medicaid and Medicare into the "Public" label, in order to mainly inspect the disparity between people receiving public and private insurance. The merging strategy follows the similar rule as in (Meng et al., 2022; Marcinkevics et al., 2022) and we acknowledge the introduced heterogeneity by this operation. However, the heterogenous group in Medicaid and Medicare are actually different manifestations of the same structural inequalities which should be considered as a whole. If a patient is covered by more than one type of health insurance in their lifetime in MIMIC-IV (which includes less than 8% of patients), we chose the health insurance type that inferred the lowest socioeconomic status for the patient (i.e. Medicaid < Medicare < Private). In the original CheXpert-Plus, the insurance type has already been simplified to as of February 2024. As for the preprocessing of other demographic attributes, the age attribute was separated into three groups: 1-39, 40-49, 50-64. The division points were established to ensure a relatively equitable distribution of patient counts across each group. As for race, we selected the two major ethnic groups in the datasets: White and Black, and the remaining ethnicities were pooled as "Others".

To decrease confounding factors that possibly affect our analysis, we only included 36,255 and 6,261 chest X-ray images in MIMIC-CXR-JPG and CheXpert, respectively, that matched the following three conditions: (1) Patient's age under 65. (2) Image labeled as "no finding" and without "support devices". (3) Frontal view image (including posteroanterior/anteroposterior views). In the first condition, we excluded patients that were automatically covered by the public insurance, and focused our analyses on economically underprivileged patients in the public insurance group; in the second condition, we

excluded patients with thoracic diseases to focus on how normal chest X-ray itself conveyed socioeconomic information; in the last condition, only frontal view images were considered to simplify our analyses. We randomly divided our extracted dataset into 0.8/0.1/0.1 splits for training, validation, and testing set, respectively, and ensured that patients in each split set did not overlap. The detailed dataset demographic distribution is shown in 1

Table 1: Dataset Demographics for MIMIC-CXR-JPG and CheXpert

**MIMIC-CXR-JPG**

| Characteristic | Train (N=29,188) | Valid (N=3,439) | Test (N=3,628) |
|---|---|---|---|
| **Gender** | | | |
| Male | 14,241 (39.0%)* | 1,675 (39.0%) | 1,840 (35.0%) |
| Female | 14,947 (36.8%) | 1,764 (36.3%) | 1,788 (36.3%) |
| **Age Group** | | | |
| 1-39 | 7,926 (42.7%) | 854 (47.0%) | 949 (43.1%) |
| 40-49 | 6,907 (37.2%) | 870 (38.7%) | 858 (36.8%) |
| 50-64 | 14,355 (35.5%) | 1,715 (32.5%) | 1,821 (31.4%) |
| **Race** | | | |
| White | 17,132 (43.5%) | 2,059 (40.1%) | 2,052 (42.4%) |
| Black | 7,220 (27.1%) | 808 (33.7%) | 935 (22.6%) |
| Others | 4,836 (33.9%) | 572 (34.6%) | 641 (33.5%) |

**CheXpert**

| Characteristic | Train (N=5,007) | Valid (N=639) | Test (N=615) |
|---|---|---|---|
| **Gender** | | | |
| Male | 2,892 (55.7%)* | 396 (58.1%) | 325 (56.0%) |
| Female | 2,115 (47.5%) | 242 (46.3%) | 290 (46.6%) |
| **Age Group** | | | |
| 1-39 | 1,886 (56.9%) | 219 (64.4%) | 229 (53.3%) |
| 40-49 | 1,086 (59.1%) | 161 (57.8%) | 154 (59.1%) |
| 50-64 | 2,035 (44.2%) | 259 (42.1%) | 232 (44.8%) |
| **Race** | | | |
| White | 2,519 (57.8%) | 322 (58.4%) | 298 (59.7%) |
| Black | 398 (24.6%) | 59 (25.4%) | 53 (22.6%) |
| Others | 2,090 (50.8%) | 258 (54.3%) | 264 (48.1%) |

* the proportion of patients with Private Insurance within that subgroup.

## 2.2. Models and Metrics

In our study, we utilized three different vision models: DenseNet121 (Huang et al., 2017), SwinV2-T (Liu et al., 2022), MedMamba (Yue and Li, 2024). Each of them represents a specific type of deep vision model.

DenseNet121 (Huang et al., 2017) comprises multiple convolutional neural network layers with denser residual connections than the traditional ResNet, which demonstrates equal

importance of density with depth and width of a model. SwinTransformer (Liu et al., 2022) is a variant of vision transformer model, where self-attention is performed on shifted image windows. This idea largely decreases the memory complexity and reduces the downsides of separated image areas. In our study, the version of SwinV2-T was utilized specifically. Lastly, MedMamba (Yue and Li, 2024) combines both a convolutional branch and a state-space sequence model branch, Mamba, to utilize the local inductive bias and global feature extraction from each branch. As for the classification, two linear layers were added after the penultimate layer of DenseNet121, SwinV2-T and MedMamba.

All of the experiments utilized Lion (Liang et al., 2024) as their optimizer, and the learning rate was 5e-6 for DenseNet121, 1e-5 for SwinV2-T and MedMamba if not otherwise mentioned. The batch size was 32 across MIMIC experiments, and 128, 128, 32, respectively for DenseNet121, SwinV2-T, MedMamba in CheXpert experiments after hyperparameter search. We did not implement dropout or weight decay for regularization for simplicity. In MIMIC-CXR-JPG, all images were resized to 448x448 otherwise mentioned, while in CheXpert, all images were resized to 320x320 since the original image size in CheXpert is smaller than in MIMIC-CXR-JPG. All images were preprocessed with RandomHorizontalFlip, RandomRotation and Normalization in PyTorch for augmentation in the training and validation process, while only Normalization was performed on testing images. A cross-entropy loss was utilized for health insurance type classification tasks. We trained our model on the training dataset, and selected the weights by the best performing average area under receiver operating characteristic curve (AUC) on the validation dataset, which is our main evaluation metric for insurance type prediction performance. The final performance was tested on the testing dataset with the chosen weights. All experiments were trained on a single NVIDIA GeForce RTX 3090, or H200 clusters.

## 3. Experiments

### 3.1. Health Insurance Prediction from CXRs

**Setting** In our first experiment, we have trained and tested our three chosen model: DenseNet121, SwinV2-T, and MedMamba on both MIMIC-CXR-JPG and CheXpert, respectively, to predict the health insurance type of each patient. We did an additional control experiment by random label perturbation (i.e. assigning label randomly to each image) on MIMIC-CXR-JPG to construct MIMIC-Random dataset.

**Results** As Table 2 indicates, in MIMIC-CXR-JPG, all trained health insurance predicting models could achieve an AUC performance over 0.61, with DenseNet having the best average AUC (0.7007) on the insurance predicting task and SwinV2-T performed the worst (0.6182). On the other hand, the models trained by CheXpert also had a better-than-random performance on the health insurance prediction, as the best performance is 0.6834. Notice that CheXpert contains less and smaller sized images compared to MIMIC-CXR-JPG, which may be the major cause of the AUC decrease in all three models. When the model was trained with the MIMIC-Random dataset, the AUC performance degraded to the level of random guess. The results supports our hypothesis that the models are not simply guessing health insurance types and that chest X-rays contain certain insurance information for the models to learn from.

Table 2: AUC Performance on Insurance Prediction

|  | DenseNet | Swin | MedMamba |
|---|---|---|---|
| MIMIC | 0.7007 | 0.6182 | 0.6684 |
| CheXpert | 0.6834 | 0.6063 | 0.6183 |
| MIMIC-Random | 0.5058 | 0.5011 | 0.4811 |

## 3.2. Localization of insurance information on Xray

**Setting** We investigated whether insurance information could be localized to a particular anatomical region or image patch with two different approaches. We divided each chest X-ray into 3x3 squared cells of approximately equal size. (Most grids are in size of 150x150, as some grids are sized 148x150 or 148x148 due to indivisible width and height.) We trained a health insurance classification model based on three vision models using two different approaches. (1) Remove-One-Patch: Select one of the nine patches, and remove all information from that patch by setting pixel values within the patch to zero. (2) Keep-One-Patch: Select one of the nine patches, and remove all information from the image by setting pixel values to zero except for the pixels within the patch. Visualization of the two approaches are shown in Figure 1 and Figure 2. MIMIC-CXR-JPG was chosen as the main dataset for this experiment due to its larger dataset size.

On the other hand, we visualized the activated image features during insurance type classification with GradCAM (Selvaraju et al., 2017) to search for localized insurance information. We generated the GradCAM visualization from one of our best-performing models - DenseNet121 at its penultimate layer on the MIMIC-CXR-JPG dataset.

**Results** Figure 1 and Figure 2 reveal the AUC performance on the insurance type prediction task for the Remove-One-Patch and Keep-One-Patch experiement. Notably there is no localized pattern in the Remove-One-Patch experiment that locates the insurance type information, and removing each patch only hurts very little performance. Conversely, there is a relatively clear localized pattern in Keep-one-patch experiment. The highest performance lands mostly in the top corners when it is the only preserved part in an image among the three models. AUC performance drops to the lowest when only the bottom one-third part of the image is retained in DenseNet121 and MedMamba. In the SwinV2-T, the patches at the bottom row also exhibit the lowest performance on average compared to the top two rows. Figure 3 also demonstrated that the highest activated features do not exhibit localization within a specific region. Combining the results, it is suggestive that the health insurance type information is diffusely distributed in the chest X-ray image (by the Remove-One-Patch experiment and GradCAM visualization), but relatively more information concentrated at the upper two-third part on the image (by the Keep-One-Patch experiement).

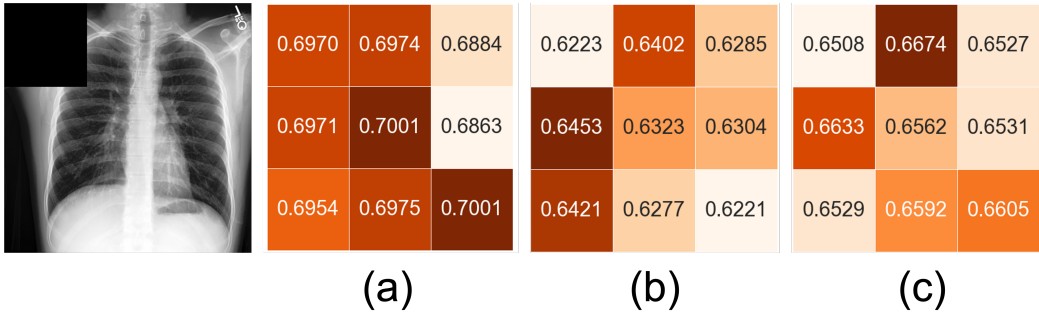

Figure 1: Spatial sensitivity of insurance type prediction to masked image regions (Remove-One-Patch). The grid images show the insurance type prediction AUC performance when masking one patch area in the 3x3 grid. The AUC figures are located at the grid position which is masked during training. (a) DenseNet121 (b) SwinV2-T (c) MedMamba

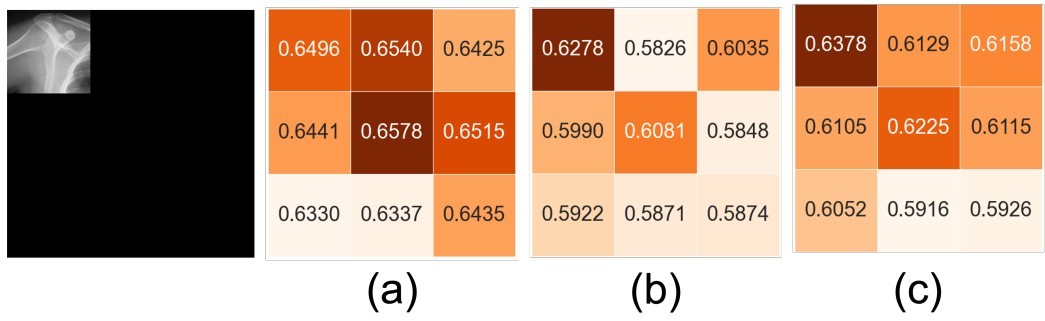

Figure 2: Spatial concentration of predictive information when only one image region is retained (Keep-One-Patch). The grid images show the insurance type prediction AUC performance when masking most of the areas in the 3x3 grid. Only one patch is retained during training. The AUC figures are located at the patch position which is retained during training. (a) DenseNet121 (b) SwinV2-T (c) MedMamba

### 3.3. Experiments on Demographic Mediators

**Setting** Based on the premise that deep learning models can effectively discern health insurance types from chest X-ray images, a pertinent inquiry arises: Is the insurance classification directly inferred, or is it mediated through demographic features like age, sex, or race? To answer this, we conducted an insurance detection experiment using other demographics. Specifically, we trained tree-based and clustering machine learning models including Random Forest (Breiman, 2001), Decision Tree (Quinlan, 1986), XGBoost (Chen

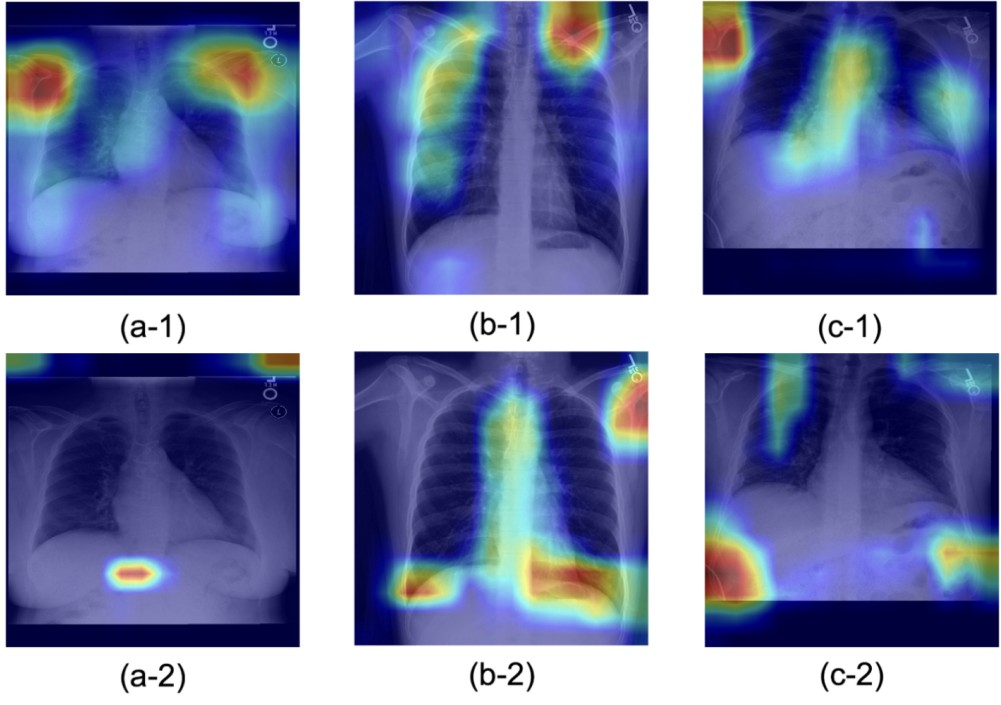

Figure 3: GradCAM result of DenseNet121 insurance prediction model on MIMIC-CXR-JPG. a, b, and c represent three different randomly chosen patients. The upper row shows the public-type insurance (Medicare, Medicaid) GradCAM visualization, while the lower row shows the private insurance GradCAM visualization.

and Guestrin, 2016), CatBoost (Prokhorenkova et al., 2018), and KNN (Fix, 1985) to predict health insurance type from a combination of demographic attributes: age, sex, and race on MIMIC-CXR-JPG. The hyperparameter is tuned by GridSearchCV from the Scikit-Learn package. In addition, given the disproportionately low private insurance coverage for Black patients compared to White patients in the dataset (1), we trained a health insurance classifier only on the White subgroup in MIMIC-CXR-JPG and CheXpert datasets. If the health insurance type prediction is mediated by the racial features, we should observe a performance degradation in the experiments. The learning rate for Lion optimizer was adjusted to 1e-4 in SwinV2-T trained with isolated White people CheXpert.

**Result** Table 3 shows the performance of multiple machine learning models to predict health insurance from the combination of age, race, and sex attributes. Among the five machine learning models (Random Forest, XGBoost, CatBoost, Decision Tree, KNN) that we have evaluated, Random Forest, CatBoost, and Decision Tree have the highest AUC value (0.6234). These results cannot explain the largely higher AUC performance in DenseNet121 and MedMamba, yet they surpass the AUC performance in SwinV2-T by about 0.005. The

lower performance in SwinV2-T may be due to the smaller size of SwinTransformer model compared to SwinV2-B and SwinV2-L, which requires further investigation. Table 4 demonstrates the health insurance type prediction performance trained solely with White people in our datasets. We observed that models trained with only White people images from MIMIC-CXR-JPG still managed to learn the health insurance information and suffers from little AUC degradation. However, the performance dropped substantially when training solely with White people in CheXpert. Notice that there is only 2508 White people in the CheXpert training dataset, compared to 17,132 White people in the MIMIC-CXR-JPG training dataset. This huge population difference provides a possible explanation for the performance degradation. On the other hand, Densenet could still maintain a high AUC performance around 0.65, suggesting that racial features should not be a significant mediator in learning health insurance type. Our results indicate that the ability of deep vision models to predict health insurance types cannot be fully explained by the mediation of demographic features.

Table 3: Health insurance type prediction performance across multiple machine learning methods given the combination of age, race, and sex attributes.

| Model | AUC |
|---|---|
| Random Forest | 0.6234 |
| XGBoost | 0.6185 |
| CatBoost | 0.6234 |
| KNN | 0.5804 |
| Decision Tree | 0.6234 |

Table 4: AUC of DenseNet121 trained on isolated White people in MIMIC-CXR-JPG and CheXpert

| | DenseNet | Swin | MedMamba |
|---|---|---|---|
| MIMIC | 0.6954 | 0.6154 | 0.6714 |
| CheXpert | 0.6535 | 0.5847 | 0.5808 |

## 4. Discussion

Interpreting these results requires considering what kinds of features AI models can extract from medical images. A useful conceptual framing distinguishes three types of features that medical-AI models can exploit: (1) biological features, directly reflecting anatomy or pathology; (2) social features, deriving from social determinants of health such as socioeconomic status or insurance types; and (3) hybrid features that intertwine both domains;

for instance, age or smoking status, which combine biological and social dimensions. The ethical and technical goal is to minimize reliance on purely social features that reflect structural disadvantage rather than biology. Simply removing explicit demographic variables is insufficient as models often reconstruct them indirectly from other data. Fairness in medical AI therefore depends not only on data curation but on recognizing and attenuating hidden social signals embedded in the pixels themselves.

Our study design makes this challenge unusually visible. We trained and evaluated models solely on radiographs labeled "no finding," images that lacked any CheXpert/MIMIC-defined abnormalities (cardiomegaly, enlarged cardiomediastinum, pulmonary edema, consolidation, pneumonia, pleural effusion, pneumothorax, atelectasis, fracture, lung lesion or opacity, and pleural or device-related findings), ensuring that all images were radiographically normal, frontal in view, and from patients under 65 years old. This design precludes shortcut learning from obvious pathology: the model could not rely on disease-related cues that might correlate with insurance status, yet its performance remained well above chance. That persistence implies the model captured either minute physiologic correlates of social conditions or contextual artifacts introduced during image acquisition. Some of these contextual artifacts may arise from system-level differences (such as hospital site, imaging hardware, or acquisition protocols) that correlate with insurance coverage, meaning the model may be detecting institutional rather than biological patterns. However, our experiments showed the strongest results using the single-site MIMIC dataset, indicating that other factors contribute to this phenomenon. The remaining signal likely originates in patient-level physiology itself, shaped over time by social environment, nutrition, stress, and comorbidity.

To examine where such subtle information might reside, we turned to spatial localization maps. Predictive regions were concentrated in the upper and mid-thorax, with minimal contribution from the lung bases. This pattern is unlikely to arise from artifacts such as tubes, leads, or devices, and may instead reflect subtle physiologic correlates. The upper and mid-thorax encompass the heart, great vessels, ribs, spine, and soft tissues; structures that may encode subtle, population-level differences linked to socioeconomic context. Variations in bone density, soft-tissue distribution, or vascular morphology shaped by chronic stress, nutrition, and healthcare access could all leave faint but detectable traces. The model's focus on these regions aligns with prior work suggesting that AI systems can infer age, sex, and even mortality risk from texture and posture cues invisible to the human eye (Ieki et al., 2022; Raghu et al., 2021; Li et al., 2022). Nevertheless, imaging-acquisition factors (e.g., variations in positioning, exposure, or calibration that correlate with insurance type) remain plausible contributors. Disentangling these biological and contextual pathways will require controlled studies linking quantitative anatomic features and imaging parameters to socioeconomic indicators.

If these cues are truly physiologic or contextual rather than demographic, they should persist across subgroups. Indeed, in experiments restricted to White participants, performance remained nearly unchanged. In another experiment predicting health insurance solely from combined demographic features also revealed lower performance. These results suggest that demographic features is not acting as a direct proxy for insurance type in Chest X-rays. The model detects something more subtle; latent, physiologically mediated imprints of social conditions that escape coarse demographic categorization. This finding

strengthens our argument that "invisible social fingerprints" can persist even when explicit social variables are absent, challenging the assumption that models trained on ostensibly neutral data are free from social bias.

Despite our results in AI algorithms predicting insurance information from chest X-ray images, there are several limitations in our work. First, both the MIMIC-IV cohort we used and the current CheXpert-Plus dataset simplify a patient's history by reporting only one insurance type. In MIMIC-IV, we specifically selected the insurance type associated with the lowest socioeconomic status. Both approaches fail to account for possible socioeconomic transitions during a patient's lifetime. However, an analysis of the original MIMIC-IV dataset revealed that only around 8% of patients ever changed their insurance status, suggesting a limited impact by this transition factor. Second, we did not explicitly show that health insurance information biases disease prediction in deep vision training, but since humans could easily learn the association between socioeconomic status and disease distribution, we should assume it being a simple association for a model to learn from. Based on past literature discovering social biases in medical image predictions (Seyyed-Kalantari et al., 2021), this is a reasonable assumption. Third, we admit that the hyperparameters may not be best-tuned in the vision models due to computational limits, especially for the suspicious drop in SwinV2-T and MedMamba performance in isolated White people in CheXpert.

Mitigating these invisible social fingerprints is the next step we would like to work on. From the dataset perspective, we should recruit patients with more comprehensive profiles and develop a rigorous surveillance pipeline to ensure the dataset follows fair rules. These include ensuring socioeconomic factors balance between target populations and the equity of data collection process. From the model perspective, we could disentangle these social signals at a higher dimensional level, and remove those which only relate to socioeconomic factors. Many previous methods have been developed to mitigate socioeconomic disparities, but they often required known attributes (e.g. age, sex, race) (Anderson and Visweswaran, 2025; Sagawa et al., 2019; Chen et al., 2024). However, our study indicates that the unethical information are much more complicated and could not be explained by a single covariate. Finally, from a deployment perspective, a sanity check is also required to ensure that the model could be safely deployed across different geographic areas and institutes.

Our work serves as a calling to re-examine the current AI models trained on chest X-ray datasets as they may utilize socioeconomic information as shortcut feature for disease diagnosis. Moreover, we are looking forward to actively developing methods that ensure algorithms do not use these shortcut features. Although many features correlated with socioeconomic status may also be legitimate biological markers of a disease, we are aiming for techniques that can disentangle these signals and completely remove the influence of features that only predict socioeconomic status, while downgrading the contribution of features that are associated with both socioeconomic status and the disease of interest. In this way, we move the field toward developing truly robust and equitable clinical tools, rather than simply creating more powerful versions of biased systems.

## 5. Conclusion

We demonstrated that the state-of-the-art deep vision models learn health insurance type information from chest X-ray images, and that it is not essentially mediated by common demographic factors. Despite the convenience of AI technology, we should utilize these deep learning models more cautiously when applying them to real-world medical tasks.

## Acknowledgments

Leo Anthony Celi is funded by the National Institute of Health through DS-I Africa U54 TW012043-01 and Bridge2AI OT2OD032701, the National Science Foundation through ITEST #2148451, a grant of the Boston-Korea Innovative Research Project (RS-2024-00403047) and a grant of the Korea Health Technology R&D Project (RS-2024-00439677) through the Korea Health Industry Development Institute (KHIDI) as funded by the Ministry of Health & Welfare, Republic of Korea. Po-Chih KUo is funded by the Ministry of Science and Technology, Taiwan (114-2628-E-007-008-MY4) and U of I-UAAT Joint Research Project (113M7054).

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

## Appendix A. Dataset description

MIMIC-IV v3.0 (Johnson et al., 2024, 2023) is a large medical dataset containing over 265,000 patients' data collected at Beth Israel Deaconess Medical Center in Boston, MA, in the intensive care unit or emergency department between 2008-2022, while MIMIC-CXR-JPG (Johnson et al., 2019a,b) is an extended image dataset for MIMIC-IV v3.0, including 377,110 chest X-ray images in total.

On the other hand, CheXpert (Irvin et al., 2019) includes 224,316 chest X-rays from 65,240 patients from both in-patient and out-patient centers of Stanford Hospital between October 2002 and July 2017. The images were downsized to 390 x 320 in the downsized version. The recent CheXpert Plus paper (Chambon et al., 2024) provides additional demographic information of each patient, including their health insurance type, race, sex, and age.

