# OpenReview forum: "Algorithms Trained on Normal Chest X-rays Can Predict Health Insurance Types"
_MIDL.io/2026/Validation_Papers — MIDL 2026 - Validation Papers Poster_

### Official Review · Reviewer_Pugv · 2026-01-02

**Confidence:** 4
**Preliminary Rating:** 4
**Final Rating:** 5

**Summary:**

This paper explores the unintended ways in which artificial intelligence (AI) models, specifically deep vision models trained on chest X-rays, can reveal hidden societal biases. The study demonstrates that models such as DenseNet121, SwinV2-B, and MedMamba can predict a patient’s health insurance type, a proxy for socioeconomic status, from normal chest X-rays, with substantial accuracy. The research challenges the notion that medical images are purely biological, revealing that AI may internalize subtle traces of social inequality, like socioeconomic segregation. Through spatial analysis, the study also identifies that these signals are diffuse, not localized to specific anatomical regions, suggesting that the models may be capturing institutional biases embedded in healthcare environments. These findings emphasize the need for AI systems in healthcare to consider the social context in which they are deployed, not just biological data, and call for deeper investigation into fairness within medical AI systems.

**Strengths:**

The study addresses a critical and under-explored issue in AI fairness within the medical domain by revealing that AI models, trained on normal chest X-rays, can predict a patient’s health insurance type — an indirect indicator of socioeconomic status. This insight exposes hidden biases embedded within the data, challenging the perception of medical images as neutral. The experimental design is robust, using multiple datasets (MIMIC-CXR-JPG and CheXpert), employing state-of-the-art deep learning models (DenseNet121, SwinV2-B*, MedMamba), and testing the hypothesis with careful attention to confounding factors like age, race, and sex. The spatial analysis via patch-based training provides valuable insights into the areas where the socioeconomic features are most concentrated within the X-rays. This research expands the scope of fairness in AI by suggesting that it’s not just about balancing datasets but understanding and addressing the social fingerprints embedded in clinical data. The paper's potential for influencing both AI ethics and medical practice is significant.

**Weaknesses:**

One notable weakness is that the study does not explicitly examine how the detected insurance-related signals influence disease predictions, which could have further validated the argument about biases in disease diagnosis. While the study controls for demographic factors, more detailed exploration of the ethical implications of using AI systems that inherently capture these "invisible social fingerprints" could deepen the discussion. Additionally, some of the results (such as the drop in performance when models were trained on a smaller dataset like CheXpert) could benefit from further clarification on why dataset size influences results more significantly in certain models. Finally, the claim that the models detect "socioeconomic segregation" through subtle traces of clinical environments could be better supported with more direct evidence linking these environmental factors to the specific X-ray features detected by the models.

**Detailed Comments:**

1. The study's design is rigorous, but it could further investigate the impact of social determinants like income, housing, and access to healthcare facilities on the AI’s predictions.

2. Further exploration of the potential medical consequences of using AI systems trained on biased data could increase the societal relevance of this paper.

3. The localization of insurance type information in specific image patches is fascinating, but a more detailed analysis of how these regions relate to potential diagnostic outcomes would be beneficial.

**Justification Of Final Rating:**

The authors have provided a thorough and well-structured response to the concerns raised in the initial review, addressing both the evidence of AI model bias and the mitigation strategies for socioeconomic disparities in medical AI.

Evidence on Socioeconomic Status Influence: The authors referenced two relevant studies (Seyyed-Kalantari et al., 2020 and 2021), which support their claim that AI models can exhibit biases across different health insurance subgroups, indirectly correlating with socioeconomic status. This additional reference strengthens their argument and provides evidence to back their claims. They have also clarified how their work explores the underlying mechanisms contributing to disparities, offering novel insights into how AI models might be unintentionally biased by diffuse signals within medical data.

Mitigation Strategies: The authors propose a multi-faceted approach to mitigate "invisible social fingerprints," which includes collecting more comprehensive datasets, developing equitable data collection practices, and employing advanced techniques like disentangling social signals in high-dimensional embedding spaces. These proposed solutions are practical and aligned with the latest advancements in fairness research in AI. Furthermore, they acknowledge that the mitigation of these biases is an ongoing challenge and suggest next steps that can be explored in future work.

The authors’ response shows a strong understanding of both the technical and ethical challenges involved in medical AI fairness. The proposed solutions are comprehensive and demonstrate an actionable path forward. Therefore, the rebuttal successfully addresses the key concerns raised in the initial review and provides a robust foundation for the final paper.

**Justification Of The Preliminary Rating:**

The paper presents novel and important findings that align with ongoing efforts to improve fairness in AI. However, further clarification and additional analysis would enhance the depth and potential impact of the study.

**Questions To Address In The Rebuttal:**

1. Could you provide further evidence or references to show that the AI models' predicted socioeconomic status influences medical outcomes, especially for underprivileged groups?

2. How do you suggest mitigating these "invisible social fingerprints" in future medical AI systems to avoid exacerbating healthcare inequalities?

---

### Official Review · Reviewer_VBZA · 2026-01-08

**Confidence:** 4
**Preliminary Rating:** 4

**Summary:**

This study investigate a critical issue in clinical ethics: whether deep learning models can predict a patient's health insurance type,which serves as a proxy for socioeconomic status, from normal chest X-rays. Using two large datasets, MIMIC-CXR-JPG and CheXpert , the authors evaluate three architectures: DenseNet121, SwinV2-B, and MedMamba. The study finds that models achieve an AUC of approximately 0.70 even after controlling for age and excluding pathological features. Further ablation experiments indicate that this signal is not mediated solely by age, sex, or race and is primarily distributed in the upper and mid-thoracic regions. This study confirm that AI can extract socioeconomic status directly from anatomically normal medical images, offering a new ethical perspective on fairness in medical AI.

**Strengths:**

The study sharply identifies that AI models may exploit "health insurance type" as a shortcut feature. The authors emphasize that if such models are deployed in clinical settings, this implicit bias embedded in medical images could lead to misdiagnosis or underdiagnosis for low-income populations. This aligns perfectly with the Special Track's pursuit of "greater clinical utility" and the prevention of potential harm from AI systems.

The authors established extremely strict inclusion criteria: patients under 65 (to exclude interference from automatic Medicare enrollment) and images labeled "No Finding" (to exclude disease severity as a confounder). This rigorous "negative design" strongly demonstrates that the signal stems from image artifacts or sub-clinical physiological traits rather than the disease itself.

The data preprocessing pipeline is described in detail, including specific screening and exclusion criteria . Key hyperparameters, such as learning rate, the use of the Lion optimizer, and batch size, are transparently disclosed . If the code link is provided in the final version as promised, reproducibility should be high.

**Weaknesses:**

While the revealed risks are valuable, the practical utility is currently limited by interpretability. The 3×3 patch-based occlusion method used is too coarse. This low-resolution analysis suggests the signal is in the "upper and mid-thoracic regions" but cannot distinguish whether the predictive features are anatomical structures (e.g., bone density, soft tissue) or external artifacts (e.g., leads, clothing). This severely limits clinicians' ability to understand what the model is actually "seeing," thereby reducing its clinical utility.

The study primarily compares AUC values based on single experiments and does not provide 95% CI or P-values. In a track that emphasizes "rigorous evaluation," this is a significant omission, making it difficult for readers to judge whether the performance differences between models are statistically significant.

The authors mention simplifying the training process by not implementing dropout or weight decay and choosing the newer Lion optimizer. The lack of stability analysis or ablation studies for these specific training strategies may introduce uncertainty for other researchers attempting to reproduce the results under different configurations.

**Detailed Comments:**

There is a risk of label heterogeneity. The authors merged Medicaid and Medicare (under 65) into a single "Public" group. However, for the population under 65, qualifying for Medicare typically implies severe disability or End-Stage Renal Disease (ESRD), whereas Medicaid is largely income-based. These two groups likely have vast physiological differences. Merging such highly heterogeneous groups suggests the model might be learning "disability features" rather than pure "socioeconomic status." The study lacks necessary stratified discussion on this issue.

First, model performance drops significantly on CheXpert, with advanced architectures like SwinV2-B and MedMamba falling to an AUC of ~0.60–0.61, while DenseNet maintains ~0.68. The authors' attribution of this solely to smaller input image size is unconvincing; further experiments are recommended to verify this. Second, the patch-based occlusion experiment was conducted only on the MedMamba architecture. To prove that the "socioeconomic signal is diffusely distributed" is a robust finding, similar trends should be replicated on DenseNet or Swin to rule out interference from architecture-specific attention patterns.

**Justification Of The Preliminary Rating:**

The study goes beyond simple image prediction. It also attempts to use demographic features (age, sex, race) for machine learning prediction as a control and performs training on White-only data to analyze the signal source from multiple angles.

The study validates its findings on two independent, recognized datasets and covers three representative deep vision architectures: CNN (DenseNet121), Transformer (SwinV2-B), and State-Space Model (MedMamba). This cross-architecture evaluation enhances the generalizability of the conclusions.

**Questions To Address In The Rebuttal:**

What exactly is the model learning?
Can the authors provide finer-grained or alternative interpretability analyses to clarify whether the socioeconomic signal arises from anatomical features (e.g., bone density, body composition) or non-biological artifacts (e.g., devices, clothing, positioning), especially given the coarse resolution of the current patch-based occlusion results?

Is “public insurance” a clean proxy for socioeconomic status?
How do the authors address the heterogeneity introduced by merging Medicaid and Medicare (under 65), where Medicare eligibility often reflects severe disability rather than income level? Have stratified or subgroup analyses been considered to disentangle disability-related signals from socioeconomic effects?

---

### Official Review · Reviewer_J9SK · 2026-01-09

**Confidence:** 3
**Preliminary Rating:** 4

**Summary:**

This paper investigates whether “normal” frontal chest radiographs—after strict filtering to remove obvious clinical and device-related cues—still contain latent information associated with socioeconomic factors, using insurance type as a proxy label. The study frames the problem as a validation task and evaluates multiple modern vision backbones on two large chest X-ray datasets. Beyond overall predictability, the paper explores where the signal may reside through occlusion-style localization analyses and considers demographic correlates as potential mediators. The findings highlight a meaningful fairness and shortcut-learning concern: even images deemed “normal” may encode non-clinical attributes that models can exploit.

**Strengths:**

High-impact, well-posed question with concrete evidence. The paper explicitly tests whether “normal” chest X-rays still carry latent socioeconomic proxies by predicting health insurance type (Public vs Private), and reports AUC ≈ 0.70 on MIMIC-CXR-JPG and ≈ 0.68 on CheXpert in the abstract, framing the finding as “invisible social signatures” in pixels rather than intended clinical content.


Cohort construction is not generic—it's deliberately confound-suppressing and clearly stated. In Sec. 2.1 (Datasets), they (i) restrict to age < 65 to avoid automatic Medicare enrollment, (ii) include only images labeled “no finding” and exclude “support devices,” and (iii) keep only frontal (PA/AP) views; this reduces the cohorts to 36,255 (MIMIC) and 6,261 (CheXpert) images and uses a patient-level 0.8/0.1/0.1 split to prevent leakage.


Architecture breadth + two datasets strengthens generality (not a one-model artifact). In Sec. 2.2, they evaluate three distinct backbone families—*DenseNet121 (CNN), SwinV2-B (ViT), and MedMamba (conv + SSM/Mamba hybrid)**—and run the insurance prediction on both MIMIC-CXR-JPG and CheXpert, using AUC as the primary metric with consistent training/selection protocol.

Sanity check is explicit and quantitatively persuasive. In Sec. 3.1, they construct MIMIC-Random via random label perturbation; the resulting AUC collapses to ~0.5 (e.g., 0.5011/0.4811/0.5058 across the three models), supporting that the signal is not merely implementation noise or training quirks.

Interpretability/localization probe is concretely designed (and the conclusion is nuanced). In Sec. 3.2, they use a 3×3 grid occlusion protocol with both Remove-One-Patch and Keep-One-Patch training/evaluation. The key result is not a single hotspot: removing any one patch hurts only slightly (diffuse information), while keeping only one patch shows a clearer gradient where upper two-thirds preserve more predictive power—supporting “diffuse but top-heavy” spatial distribution.


Mediator analysis is tied to concrete baselines and subgroup training, not hand-waving. In Sec. 3.3, they test whether insurance prediction is mediated by demographics via (i) classical ML models trained on age+sex+race (best AUC only ~0.59), and (ii) training vision models on an isolated White subgroup, where performance remains close on MIMIC (limited degradation) and drops more on CheXpert with an explicit sample-size caveat—supporting the claim that coarse demographics are insufficient to explain the image signal.


Practice relevance is explicitly argued in Discussion/Conclusion, with a clear “so what.” The Discussion frames “biological vs social vs hybrid features” and argues that removing explicit demographics is insufficient because models can reconstruct social signals from pixels; the Conclusion calls for more cautious deployment and motivates methods to disentangle/remove socioeconomic-only signals.

**Weaknesses:**

The paper does not yet pin down what the models are using to predict insurance. While the restrictive cohort construction is a strong start (Sec. 2.1), the current experiments do not directly control for common acquisition/workflow correlates (e.g., AP/PA differences, portable imaging, scanner/site artifacts, borders/markers). Since the discussion already acknowledges institutional/care-pathway effects, adding targeted controls would make the fairness interpretation substantially stronger.

The insurance label definition could influence the results but is not stress-tested. In Sec. 2.1, multi-insurance patients are handled with a fixed mapping rule (favoring the “lowest SES” insurance). A small sensitivity analysis (e.g., using insurance at imaging time, or excluding multi-insurance patients) would clarify whether the main conclusions are robust to this choice.

The localization evidence is suggestive but may depend on the perturbation method. The 3×3 patch occlusion design (Sec. 3.2) is helpful as a first probe, but zeroing patches can introduce unnatural artifacts. Corroborating with a less artifact-prone perturbation (blur/inpainting) or an anatomy-aware test (lung-only vs non-lung) would make the “where is the signal” claim more convincing.

Robustness reporting could better match the fairness motivation. The demographic mediator analysis (Sec. 3.3) is a good step, but the paper would benefit from clearer stability evidence (multiple seeds/splits, and subgroup consistency beyond a single restricted subset) to support deployment-relevant conclusions.

**Detailed Comments:**

This paper presents a careful validation study showing that deep vision models can predict health insurance type from radiographically normal chest X-rays. The experimental design is generally sound, particularly the restriction to frontal-view, “no finding” images and patients under 65, which meaningfully reduces several obvious confounders.

A few minor points could improve clarity:

The manuscript would benefit from more explicit discussion of what conclusions cannot be drawn from the current experiments, especially regarding the distinction between physiological signals and acquisition- or workflow-related artifacts. While both possibilities are mentioned, the boundary between them is sometimes implicit.

In Section 3.2, the patch-based experiments support the claim of diffuse signal distribution. Clarifying that the 3×3 grid is a coarse probe rather than a precise localization method would help set appropriate expectations for interpretation.

The choice to merge Medicaid and Medicare into a single “Public” category is reasonable for the stated goal, but a short clarification in the Methods explaining how this affects interpretability would improve transparency.

**Justification Of The Preliminary Rating:**

This paper provides a solid and well-controlled validation study demonstrating that modern vision models can predict health insurance type from normal chest X-rays. The contribution is not a new model or method, but rather a carefully designed empirical finding that challenges common assumptions about the neutrality of medical imaging data.

The authors take meaningful steps to reduce confounding, including restricting to “no finding” images, excluding patients over 65, and analyzing demographic mediators. The consistency of results across architectures and datasets strengthens the credibility of the core claim. While the precise origin of the signal remains unresolved, the paper is appropriately cautious in its interpretation and frames its conclusions within the limits of the presented evidence.

Overall, this work fits well within the scope of MIDL as a validation and problem-framing paper and should be of interest to the community, even though mechanistic explanations remain an open question.

**Questions To Address In The Rebuttal:**

The paper discusses both physiological correlates and acquisition or institutional factors as potential sources of the insurance signal. Based on the authors’ analyses, can they clarify which explanations they view as better supported by the current evidence, and which remain speculative?

For patients with multiple insurance types in MIMIC-IV, the study assigns the lowest socioeconomic-status label. Do the authors expect the main conclusions to change if alternative labeling rules were applied, and why or why not?

In Section 3.3, demographic-only models achieve AUCs below 0.6. How should readers interpret this result in relation to the imaging models’ performance—does it primarily rule out direct mediation, or could weaker indirect mediation still be present?

---

### Meta-Review · Area_Chair_VipH · 2026-02-04

**Recommendation:** Accept (Poster)
**Confidence:** 4

**Metareview:**

All reviewers found the paper to be a solid, well-designed validation study on predicting health insurance type from standard chest X-rays. The experimental methodology is rigorous, and the empirical evidence is presented clearly. Overall, the proposed approach is comprehensive and practical, outlining an actionable path for follow-up work and potential downstream deployment.

---

### Decision · Program_Chairs · 2026-02-14

Accept (Poster)